# Adaptive Adjacent Layer Feature Fusion for Object Detection in Remote Sensing Images

**Xuesong Zhang, Zhihui Gong, Haitao Guo, Xiangyun Liu \*, Lei Ding** **, Kun Zhu**  **and Jiaqi Wang**

Institute of Geospatial Information, PLA Strategic Support Force Information Engineering University, Zhengzhou 450001, China; zxs_2023@163.com (X.Z.)

\* Correspondence: liu_xy1994@163.com

**Abstract:** Object detection in remote sensing images faces the challenges of a complex background, large object size variations, and high inter-class similarity. To address these problems, we propose an adaptive adjacent layer feature fusion (AALFF) method, which is developed on the basis of RTMDet. Specifically, the AALFF method incorporates an adjacent layer feature fusion enhancement (ALFFE) module, designed to capture high-level semantic information and accurately locate object spatial positions. ALFFE also effectively preserves small objects by fusing adjacent layer features and employs involution to aggregate contextual information in a wide spatial range for object essential features extraction in complex backgrounds. Additionally, the adaptive spatial feature fusion (ASFF) module is introduced to guide the network to select and fuse the crucial features to improve the adaptability to objects with different sizes. The proposed method achieves mean average precision (mAP) values of 77.1%, 88.9%, and 95.7% on the DIOR, HRRSD, and NWPU VHR-10 datasets, respectively. Notably, our approach achieves $mAP_{75}$ values of 60.8% and 79.0% on the DIOR and HRRSD datasets, respectively, surpassing the state-of-the-art performance on the DIOR dataset.

**Keywords:** adjacent layer feature; object detection; remote sensing image

## 1. Introduction

Object detection in remote sensing images is the process of positioning and recognition of interested objects. It has a wide range of applications, such as intelligent detection [1], port management [2], military reconnaissance [3], urban planning [4], etc.

Traditional object detection methods [5,6] need to design features according to the characteristics of the interested objects, and have problems such as low efficiency and poor generalization ability, which cannot meet the requirements in practical applications.

In recent years, the advancement of deep learning techniques has been remarkable, with convolutional neural networks (CNNs) emerging as a dominant approach in various image recognition tasks [7–9]. This is primarily attributed to their remarkable ability to extract and represent features effectively.

Currently, object detection methods based on CNNs can be categorized into two-stage and one-stage approaches. Two-stage methods initially generate candidate regions, filtering out numerous background boxes, and balancing positive and negative samples to enhance object detection recall. Subsequently, the objects of interest are classified and localized based on this analysis. These methods, such as Fast R-CNN [10] and Faster R-CNN [11], are recognized for their higher accuracy but lower efficiency. On the other hand, one-stage object detection methods directly estimate object positions and categories from input images, eliminating the need for additional candidate region generation. These methods are characterized by their simpler and more efficient structures, enabling fast real-time detection in various applications. Examples of one-stage methods include YOLO [12], SSD [13], and RetinaNet [14].

However, in different imaging platforms and imaging methods, the objects in optical remote sensing images may appear to have different aspect ratios, spatial deformation,

orientations, and spatial resolutions. Different methods have been developed to address these problems. Yu et al. [15] employed deformable convolution to align feature maps of different scales, and designed a feature fusion module using dilated convolution to enhance the perception of object shape and direction. Hou et al. [16] designed an asymmetric feature pyramid network to enrich the spatial representation of features and improve the detection of objects with extreme aspect ratios. Wang et al. [17] combined pooling and dilated convolution to design a global feature information complementary module, which aggregated non-neighborhood multi-scale contextual information while enhancing the detailed feature information of the object to improve the model's detection of objects at different scales.

Although progress has been made in object detection of remote sensing images, challenges still remain. These challenges include the following:

(1) Remote sensing images have a broader coverage range and more complex backgrounds compared to natural scene images, which severely interfere with object detection. Extracting discriminative features of the objects becomes difficult, as illustrated in Figure 1a, where a train station appears similar to the surrounding buildings. Moreover, backgrounds that resemble the texture or shape of objects can lead to false detections.

(2) Some object categories exhibit high similarity, such as the basketball courts and tennis courts shown in Figure 1b which share similar appearance features. To differentiate between them, it is necessary to utilize texture or contextual information as an auxiliary means.

(3) Objects exhibit significant scale variations, both the inter-class and intra-class objects. Figure 1c demonstrates airplanes of different scales, while Figure 1d depicts vehicles and overpasses with considerable scale differences. These variations undoubtedly increase the difficulty of accurate detection.

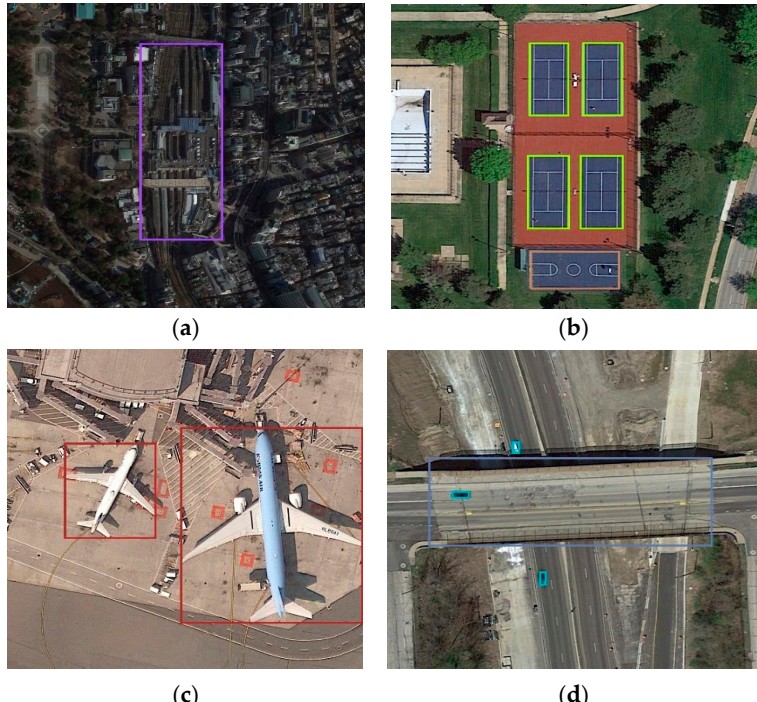

**Figure 1.** Examples of the aforementioned challenges. (**a**) A train station amidst a complex background; (**b**) visually similar basketball and tennis courts; (**c**) airplanes with significant scale variations, and (**d**) vehicles and overpasses with considerable scale differences (The images are sourced from the DIOR dataset).

To address the aforementioned challenges, we propose an adaptive adjacent layer feature fusion object detection method based on RTMDet [18], focusing on the fusion interaction of highly correlated adjacent layer features and the adaptive fusion of different scale features, so as to capture the discriminative features of objects. The main contributions of this paper are as follows:

(1) An adjacent layer feature fusion enhancement (ALFFE) module is designed to enable the model to embed the discriminative spatial and semantic information. ALFFE fuses adjacent layer features to improve spatial perception and capture high-level object semantics. Involution is employed to overcome the difficulty of extracting object semantic information in complex backgrounds.

(2) In order to make full use of features at different scales, the adaptive spatial feature fusion module (ASFF) is introduced to improve the scale invariance of the features and better adapt to objects of different scales by adaptively fusing the multi-scale features at each spatial location.

(3) Extensive experiments on multiple datasets demonstrate the effectiveness of our method, with state-of-the-art accuracy achieved on the DIOR dataset.

## 2. Related Work

### 2.1. One-Stage Object Detection Methods

One-stage object detection methods have gained prominence due to their balance between speed and accuracy. These methods can be further classified into anchor-based and anchor-free approaches, depending on whether they employ predefined anchors.

Anchor-based methods generate a large number of anchors across an image to improve recall and precision. Examples of anchor-based methods are YOLOV3 [19], YOLOV4 [20], SSD, and RetinaNet. However, this approach often leads to an imbalance between positive and negative samples, as only a few anchors effectively cover the objects of interest. Furthermore, if the scale and size of the anchors significantly differ from those of the objects, detection accuracy may decrease. For instance, remote sensing images containing objects with large aspect ratios such as ships, airports, and bridges may experience challenges in accurate detection if the anchors fail to adequately cover them.

To mitigate the impact of anchor settings, researchers have explored anchor-free object detection methods that eliminate the need for predefined anchors. These methods primarily rely on predicting key points or center points of objects. CornerNet [21] transforms the object bounding box detection problem into key point detection, specifically targeting the top-left and bottom-right vertices of the objects. By establishing correspondences between these key points, the boundaries of the objects are determined. CenterNet [22] treats objects as single points and predicts their center points, while FCOS [23] transforms anchors into anchor points by predicting the distances between anchor points and the four edges of object bounding boxes. These anchor-free methods exhibit promising potential. Recent notable methods such as YOLOX [24] and YOLOV6 [25] have emerged, further advancing the anchor-free approaches.

In this paper, the anchor-free method RTMDet (real-time models for object detection) is used as the baseline. The overall structure of RTMDet is similar to that of YOLOX, consisting of CSPNeXt, CSPNeXtPAFPN, and SepBNHead. The internal core module of CSPNeXt is the improved CSPLayer [26], where the original basic block (Figure 2a) is refined utilizing a $5 \times 5$ depthwise separable convolution (Figure 2b). The $5 \times 5$ depthwise separable convolution is used to increase the receptive field and improve the feature extraction capability at a small computational cost. A channel attention module is also integrated into the CSPLayer to further improve the performance of the model via learning to assign appropriate weights to channels, enabling the model to better capture and utilize discriminative features. The CSPNeXtPAFPN uses the same basic building blocks as CSPNeXt with bottom-up and top-down feature propagation [27]. In the detection heads, RTMDet utilizes SepBNHeads, which share convolution weights and separate batch normalization layers, since RTMDet believes that the detected features of the object are similar for different

feature levels at relative scale sizes and introducing BN directly into the detection heads with shared parameters would result in sliding means and lead to error.

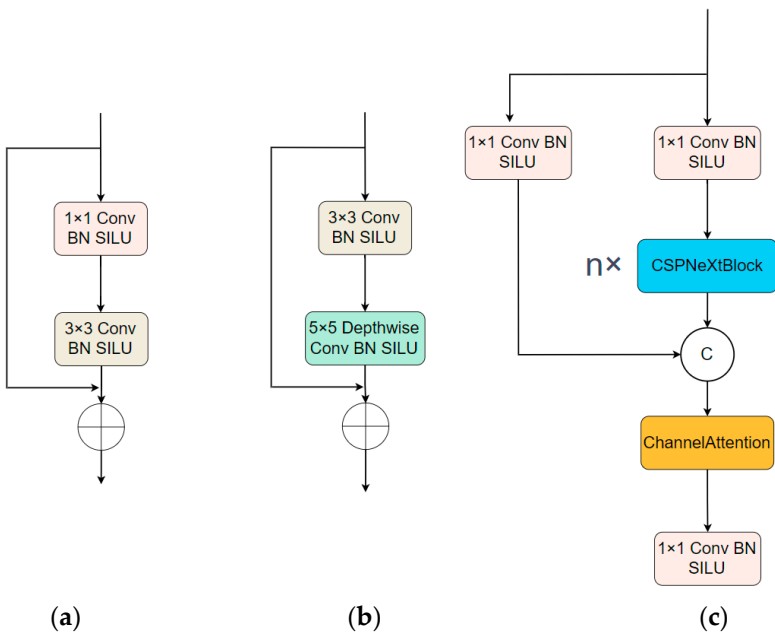

**Figure 2.** Basic block and improved CSPLayer. (**a**) CSPDarknet basic block; (**b**) CSPNeXt basic block; (**c**) improved CSPLayer.

CSPNeXt is divided into five different sizes according to the depth, such as tiny, s, m, l, and x. In this paper, we choose CSPNeXt-m with moderate accuracy and complexity. CSPNeXt-m can output five features with different resolutions noted as $C_1$, $C_2$, $C_3$, $C_4$, and $C_5$, respectively. $C_3$, $C_4$, and $C_5$ are used for subsequent feature fusion enhancement since $C_1$ and $C_2$ reside in the shallow layer of the network and possess limited spatial and contextual information. Considering an input image size of $H \times W \times 3$, the size of the output features $C_3$, $C_4$, and $C_5$ are $H/8 \times W/8 \times 192$, $H/16 \times W/16 \times 384$, and $H/32 \times W/32 \times 769$.

### 2.2. Multi-Scale Feature Fusion

Remote sensing images usually contain various types of objects, which may vary greatly in scale, and the size of objects in the same category also varies in scale depending on the image resolution. In addition, there are numerous small objects in remote sensing images, and for small objects it is easy to lose information in the feature extraction down-sampling process, resulting in missing detection. These problems bring great challenges for the task of object detection in remote sensing images. The multi-scale feature fusion method, which combines objects information at multiple scales for judgement, is an effective method to solve these problems. At present, many researchers have performed research in this field. Li et al. [28] obtained richer context information using bidirectional fusion of deep features and shallow features as well as skip connections. Dong et al. [29] improved the detection performance of remote sensing objects at different shapes and sizes by aggregating the multilevel output of FPN [30] and the global contextual information of the entire image. Li et al. [31] used a dual-path structure and deformable convolution to improve the feature extraction capability for rotating objects, and utilized channel attention and adaptive feature fusion to guide the feature layers to learn and retain the most appropriate features. Xu et al. [32] alleviated the difficulty of extracting object semantic information due to changes in contextual information by fusing original and complementary features. Wang et al. [33] proposed a feature reflow pyramid structure to improve feature representation through adding a feature flow path from a lower level to each scale. Lv et al. [34] integrated

features from three different scales into features at the same scale, while using a spatial attention weighting module to adjust the fusion ratio of features at different scales to enhance the flexibility of the fusion method. Cheng et al. [35] inserted channel attention into the top level of the feature pyramid network to construct the relationship between the feature maps of different channels, and obtained more discriminative multilevel features by fusing features of different scales.

The method proposed in this paper focuses on the fusion and interaction of the features of adjacent layer with high correlation, and further mining and exploiting the contextual information to enhance feature representation. Furthermore, in order to make full use of multi-scale features, ASFF [36] is employed to adaptively adjust the degree of respective involvement of multi-scale features in the fusion process to improve the scale invariance of the features.

## 3. Methodology

### 3.1. Overview

The proposed method is developed on the framework of the anchor-free detector RTMDet [18], which consists of three parts: feature extraction, adaptive adjacent layer feature fusion, and SepBNHeads. The architecture of the proposed method is shown in Figure 3. In the feature extraction, three different scales of the image features are extracted using CSPNeXt-m and are noted as $C_3$, $C_4$, and $C_5$, respectively. In the adaptive adjacent layer feature fusion, the features dimensions are aligned, and the deep layer features are upsampled and fused with the adjacent layer features to obtain the features $F_3$, $F_2$, and $F_1$, respectively. Then, adjacent layer features are deeply fused by ALFFE to enhance the feature representation. Finally, ASFF is employed to enhance the scale invariance and discrimination of features by learning adaptive weights for the fusion of three different scales of features achieving adaptive fusion. In the detection heads, we utilize SepBNHeads in line with the RTMDet.

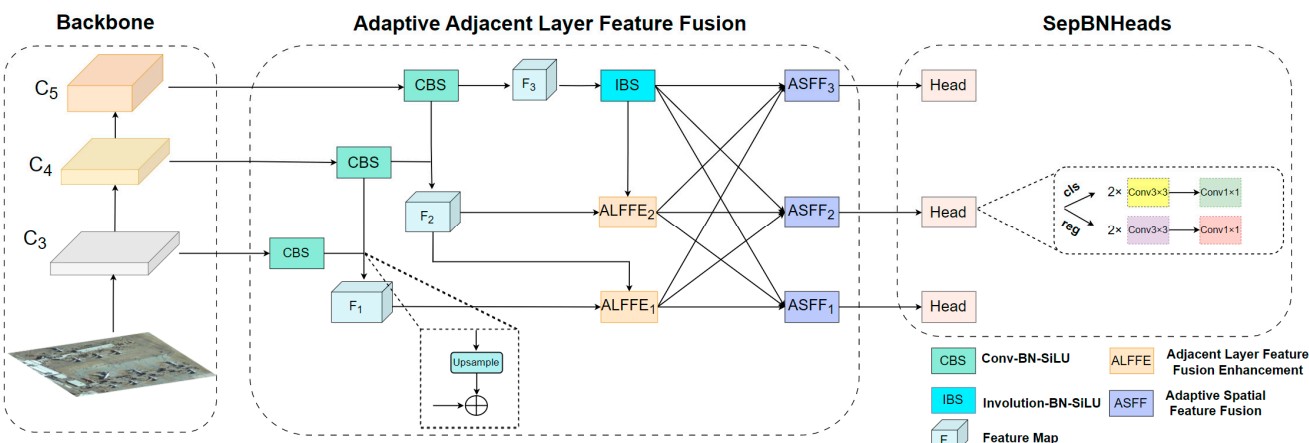

**Figure 3.** The architecture of the proposed method. The multilevel features, noted as $C_3$, $C_4$, and $C_5$, are extracted from the backbone. SepBNHeads denote decoupled detection heads with shared convolution weights and separated batch normalization layers.

### 3.2. Involution

In contrast to the spatial independence and channel specificity of convolutional operations, involution [37] is spatial-specific and channel-agnostic. Involution kernels are distinct in the spatial extent but shared across channels, which reduces the kernel redundancy. On the one hand, involution can aggregate contextual information over a wider spatial range, capturing long-distance spatial relationships with a larger receptive field. On the other hand, involution adaptively assigns weights at different locations. In other words, it assigns large weights to information-rich places in the spatial domain and focuses more on these places.

Given an input feature map $X \in R^{H \times W \times C}$ with a pixel $X_{i,j} \in R^c$, the corresponding involution kernel $\mathcal{H}_{i,j,\dots,g} \in R^{K \times K}$, $g = 1, 2, 3, \cdots, G$, is fixed. $H$, $W$, and $C$ denote the height, width, and number of channels of the feature map, respectively. $g$ denotes the number of groups sharing the same convolution kernel in the channel dimension and $K$ denotes the size of the kernel. The feature map output by involution is obtained by multiplying and adding the input feature map by the involution kernel. The definition is as follows:

$$Y_{i,j,k} = \sum_{(u,v) \in \Delta K} \mathcal{H}_{i,j,u+[\frac{K}{2}],\ [\frac{kG}{C}]} X_{i+u,j+v,k}. \tag{1}$$

The shape of the involution kernel $\mathcal{H}$ depends on the shape of the input feature map $X$. The kernel generation function is symbolized as $\phi$, and the calculation of $\mathcal{H}_{i,j}$ is as follows:

$$\mathcal{H}_{i,j} = \phi(X_{i,j}) = W_1 \sigma(W_0 X_{i,j}) \tag{2}$$

where $W_0 \in R^{\frac{C}{r} \times C}$ and $W_1 \in R^{(K \times K \times G) \times \frac{C}{r}}$ denote two linear transformations, $r$ is the channel reduction ratio, and $\sigma$ denotes the non-linear activation function for the two linear transformations after batch normalization.

### 3.3. Adjacent Layer Feature Fusion Enhancement Module

The adjacent layer features are highly correlated and do not have a "semantic gap" due to the large differences between features. Therefore, we consider fusing the features of adjacent layers further improves feature representation. The deep features correspond to a larger receptive field and contain more semantic information, which is beneficial in identifying the object category. However, they do not contain much spatial information, which is crucial to object positioning. On the contrary, shallow features contain more spatial information, such as the texture of objects, which is conducive to object localization. Meanwhile, the semantic representations are weak, which is not conducive to identifying the category of objects.

The simultaneous fulfillment of accurate object localization and class recognition requirements can be achieved by transferring and fusing information between features of adjacent layers. Moreover, small objects in images often have limited available information, and their spatial information gradually diminishes with increasing network depth, leading to missed detections. To address this, we propose an adaptive adjacent layer feature fusion enhancement (AALFFE) module. The structure of this module is illustrated in Figure 4. Firstly, involution with spatial specificity and channel independence is employed to aggregate contextual information from both deep and shallow features, enhancing the extraction of discriminative features within complex backgrounds. Next, a $1 \times 1$ convolution is used to reduce the number of channels to 128, simultaneously deepening the network and improving its modeling capabilities. Additionally, deep and shallow features are further extracted using a $5 \times 5$ depthwise separable convolution, which offers a larger effective receptive field compared to the commonly used $3 \times 3$ convolution. This enables the capture of more local spatial and contextual information with reduced computational effort. Subsequently, a $1 \times 1$ convolution is employed to increase the number of feature channels to 256, enhancing information interaction across channels. The deep features are then upsampled to match the spatial size of the adjacent shallow features, and element-wise addition is performed to synthesize the feature information from the adjacent layers, improving feature representation. Finally, the shallow features and the features obtained after initial fusion are concatenated along the dimension, and a $3 \times 3$ convolution is applied to further strengthen the fusion and interaction of spatial and channel information.

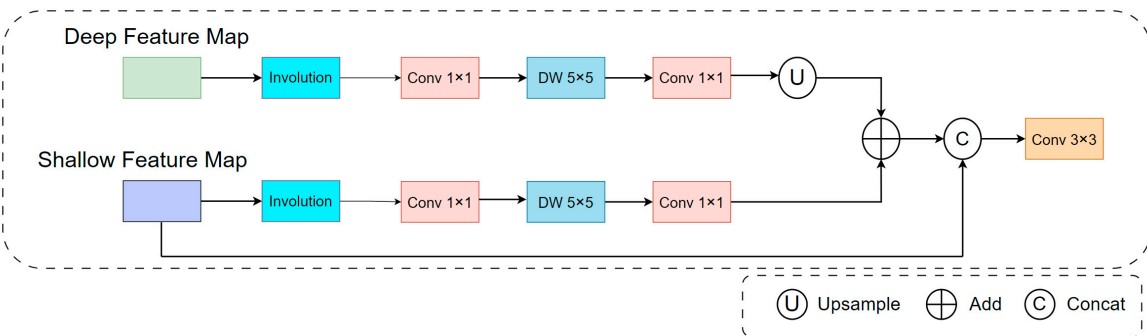

**Figure 4.** The architecture of ALFFE.

After aligning the channels number of features $C_3$, $C_4$, and $C_5$ to 256 dimensions, $C_5$ and $C_4$ are upsampled by a factor of 2 to obtain features $F_3$, $F_2$, and $F_1$ after fusion with $C_4$ and $C_3$, respectively, in the form of element-by-element addition. The process can be formulated as follows:

$$F_3 = CBS_{1\times1}(C_5) \tag{3}$$

$$F_2 = CBS_{1\times1}(C_4) + Upsample(CBS_{1\times1}(C_5)) \tag{4}$$

$$F_1 = CBS_{1\times1}(C_3) + Upsample(C_4) \tag{5}$$

Let us consider the features $F_2$ and $F_1$. The feature $F_{ALFFE_1}$ is obtained by feeding $F_2$ and $F_1$ into the adjacent layer feature fusion enhancement module. The process can be formulated as follows:

$$F_{21} = Upsample(CBS_{1\times1}(DWBS_{5\times5}(CBS_{1\times1}(IBS(F_2))))) \tag{6}$$

$$F_{11} = CBS_{1\times1}(DWBS_{5\times5}(CBS_{1\times1}(IBS(F_1)))) \tag{7}$$

$$F_{ALFFE_1} = CBS_{3\times3}(Cat(F_{21} + F_{11}, F_1)) \tag{8}$$

Similarly, $F_{ALFFE_2}$ is obtained using feature $F_3$ and feature $F_2$. The formula is as follows:

$$F_{ALFFE_2} = CBS_{3\times3}(Cat(F_{32} + F_{22}, F_2)) \tag{9}$$

In Equations (3)–(9), $CBS_{n\times n}(\cdot)$ represents a module consisting of a convolutional layer (with a kernel of size $n \times n$), a BN layer, and a SiLU layer. Similarly, $DWBS_{5\times5}(\cdot)$ denotes a depthwise separable convolution (with a kernel of size $5 \times 5$), followed by a BN layer and a SiLU layer. $IBS(\cdot)$ denotes an involution layer, followed by a BN layer and a SiLU layer. $Upsample(\cdot)$ denotes bilinear interpolation upsampling. $Cat(\cdot)$ denotes the feature concatenation operation.

### 3.4. Adaptive Spatial Feature Fusion

The ASFF module assigns weights for different levels of features, which determines the degree of involvement during the fusion. This enables the adaptive selection of features, thus improving the robustness of the feature embedding. In this process, the weights of features are determined by the global information and how favorable they are to the object.

The number of channels and size of features must be aligned before the fusion. Given the number of channels of the features are already aligned in Section 3.3, it is only required to align the sizes of the features at different scales. In addition, the extraction of semantic

information in feature $F_3$ is further enhanced by using involution before feature adaptive fusion to obtain feature $F_{33}$.

$$F_{33} = IBS(F_3) \tag{10}$$

where $IBS(\cdot)$ denotes an involution layer, a BN layer, and a SiLU Layer.

For $ASFF_1$: $X^{3\rightarrow1}$ is obtained by upsampling feature $F_{33}$ by a factor of 4, while $X^{2\rightarrow1}$ is obtained by upsampling feature $F_{ALFFE_2}$ by a factor of 2. For $ASFF_2$: $X^{3\rightarrow2}$ is obtained by convolving the feature $F_{33}$ with a $3 \times 3$ kernel and stride 2, while $X^{1\rightarrow2}$ is obtained through upsampling the feature $F_{ALFFE_1}$ by a factor of 2. For $ASFF_3$: feature $F_{ALFFE_1}$ is obtained by applying maxpooling with a $3 \times 3$ kernel and stride 2, followed by convolution with a $3 \times 3$ kernel and stride 2 to obtain feature $X^{1\rightarrow3}$. Similarly, $X^{2\rightarrow3}$ is obtained by convolving feature $F_{ALFFE_2}$ with a $3 \times 3$ kernel and stride 2. The features are adjusted according to the following equations:

$$X^{3\rightarrow1} = Up_4(F_{33}) \tag{11}$$

$$X^{2\rightarrow1} = Up_2(F_{ALFFE_2}) \tag{12}$$

$$X^{3\rightarrow2} = CBS_{3\times3}(F_{33}) \tag{13}$$

$$X^{1\rightarrow2} = Up_2(F_{ALFFE1}) \tag{14}$$

$$X^{1\rightarrow3} = CBS_{3\times3}(Maxpooling_{3\times3}(F_{ALFFE_1})) \tag{15}$$

$$X^{2\rightarrow3} = CBS_{3\times3}(F_{ALFFE_2}) \tag{16}$$

In Equations (11)–(16), $Up_4(\cdot)$ and $Up_2(\cdot)$ denote nearest interpolation upsampling by factors of 4 and 2, respectively; $CBS_{3\times3}(\cdot)$ denotes a module consisting of a convolution layer with a $3 \times 3$ kernel, a BN layer, and a SiLU layer. $Maxpooling_{3\times3}(\cdot)$ indicates maxpooling with a $3 \times 3$ kernel and stride 2.

Then, the adaptive fusion feature $ASFF_l$ is obtained by weighted addition of the three adjusted features. The vectors at any spatial location $(i, j)$ on the fused feature map are a weighted fusion of the vectors at the same spatial location on the three feature maps before fusion, with the coefficients (the spatial importance weights of the feature maps) being adaptively learned by the network and are shared across all channels. The formula is as follows:

$$ASFF_l = \alpha_{ij}^l X^{1\rightarrow l} + \beta_{ij}^l X^{2\rightarrow l} + \gamma_{ij}^l X^{3\rightarrow l} \tag{17}$$

where $\alpha_{ij}^l + \beta_{ij}^l + \gamma_{ij}^l = 1$, $\alpha_{ij}^l$, $\beta_{ij}^l$, $\gamma_{ij}^l$ are three resized feature maps obtained using the softmax formula after stitching in dimensions, and all take values in the range [0,1]. For example, the calculation of $\beta_{ij}^l$ is as follows:

$$\beta_{ij}^l = \frac{e^{\lambda_{\beta_{ij}}^l}}{e^{\lambda_{\alpha_{ij}}^l} + e^{\lambda_{\beta_{ij}}^l} + e^{\lambda_{\gamma_{ij}}^l}} \tag{18}$$

where $\lambda_{\alpha_{ij}}^l$, $\lambda_{\alpha_{ij}}^l$, and $\lambda_{\alpha_{ij}}^l$ are obtained from the adjusted features through convolution with a $3 \times 3$ kernel.

## 4. Experiment

To evaluate the performance of backbone, ALFFE, and ASFF, ablation studies are conducted on the DIOR [38] dataset and HRRSD [39] dataset to evaluate the performance of the proposed method. The performance of the proposed method is assessed on the NWPU VHR-10 [40] dataset.

### 4.1. Datasets

The DIOR dataset represents a significant contribution to the field of remote sensing image object detection, offering a comprehensive collection of 23,463 images encompassing 192,472 instances and 20 common categories. These categories span a wide range of objects, including airplane (AP), airport (AI), baseball field (BD), basketball court (BC), bridge (BR), chimney (CH), dam (DA), expressway service area (ESA), expressway toll station (ETS), harbor (HA), golf field (GF), ground track field (GTF), overpass (OV), ship (SH), stadium (SD), storage tanks (ST), tennis court (TC), train station (TS), vehicle (VE), and windmill (WM). Each category is represented by approximately 1200 images, providing ample data for training and evaluation.

The images in the DIOR dataset exhibit dimensions of $800 \times 800$ pixels, with varying resolutions, ranging from 0.5 m to 30 m. This diversity in image resolution introduces challenges related to scale variations and high inter-class similarity, further compounded by intra-class diversity. To ensure scientific rigor, the dataset provides official data divisions for experimentation, allocating 5862 images for model training, 5863 images for model validation, and 11,738 images for model testing.

In comparison, the HRRSD dataset, released in 2019, is another notable remote sensing image object detection dataset. It comprises 21,761 images featuring 55,740 object instances distributed across 13 common classes. The common categories encompass airplane (AP), baseball field (BD), basketball court (BC), bridge (BR), crossroads (CR), ground track field (GTF), harbor (HA), parking lot (PL), ship (SH), storage tank (ST), T junction (TJ), tennis court (TC), and vehicle (VE). Each category is well represented with approximately 4000 object instances, ensuring a robust and diverse dataset.

The NWPU VHR-10 dataset, on the other hand, represents a valuable resource in the realm of remote sensing image object detection. It consists of 650 annotated images, encompassing 3651 object instances across 10 common categories. These categories include airplane (AP), ship (SH), storage tank (ST), baseball field (BD), tennis court (TC), basketball court (BC), ground track field (GTF), harbor (HA), bridge (BR), and vehicle (VE). While no specific data division is provided, a random selection process is employed, allocating 60% of the images for training, 20% for validation, and the remaining 20% for testing purposes.

### 4.2. Experimental Settings and Evaluation Metrics

(1)    Implementation Details

We implement our method based on MMDetection [41], with an Intel Core i9-10900K CPU and an NVIDIA RTX3090 GPU. For experiments on the three datasets, the input image size is set to $800 \times 800$. We use the AdamW [42] optimizer for model optimization, with the weight decay set to 0.05. The feature extraction network CSPNeXt-m is initialized using the weights pretrained in ImageNet. The initial learning rates are set to 0.004, 0.002, and 0.01 on the DIOR, HRRSD, and NWPU VHR-10 datasets, respectively. The batch sizes are set to 8, 8, and 6, respectively. The learning rates are tuned using a combined QuadraticWarmupLR and CosineAnnealingLR strategy. On the DIOR and HRRSD datasets, experiments are conducted using a two-stage training strategy and mixed precision accelerated training for a total of 300 epochs. Data enhancements such as Mosic, Mixup, and random flip are used in the first 270 epochs of training to improve the model performance. However, the Mosic and Mixup enhancements are extremely distorted, so weak data enhancements such as random resize and flip are used in the second stage. In the last 30 epochs, weak data enhancements are used to allow the model to be fine-tuned in a state that better matches the feature distribution of the original dataset. On the NWPU VHR-10 dataset, a total of 160 epochs are trained without mixed precision training, with data enhancements such as random resize and flip.

(2)    Evaluation Metrics

In the conducted experiments, the mean average precision (mAP), frames per second (FPS), and the number of parameters are employed as performance evaluation metrics

to comprehensively assess the model's performance. The mAP represents the average precision (AP) across all categories and serves as a comprehensive metric commonly used for evaluating object detection model performance. The AP value for each category is calculated as the area under the precision–recall (PR) curve, plotted with recall on the x-axis and precision on the y-axis.

Precision refers to the proportion of samples correctly classified as positive samples out of all samples detected as positive. Recall, on the other hand, represents the proportion of positive samples that are correctly detected. Precisely, precision and recall are defined as follows:

$$Precision = \frac{TP}{TP + FP} \tag{19}$$

$$Recall = \frac{TP}{TP + FN} \tag{20}$$

where $TP$ denotes the number of correctly detected objects, $FP$ denotes the number of incorrectly detected objects, and $FN$ denotes the number of missed objects.

The $AP$ and $mAP$ can be calculated using the following formulas:

$$AP = \int_0^1 P(R)dR \tag{21}$$

$$mAP = \frac{1}{K} \sum_{i=1}^{K} AP^i \tag{22}$$

where $K$ denotes the number of categories, and larger values of $AP$ and $mAP$ denote more accurate prediction results.

### 4.3. Ablation Study

In order to validate the effectiveness of ALFFE and ASFF of the proposed method, ablation experiments are conducted on the DIOR dataset and the HRRSD dataset. Tables 1 and 2 present the achieved mean average precision (mAP) values using different methods on the DIOR and HRRSD datasets, respectively. Here, $mAP_{50}$ and $mAP_{75}$ represent the mAP values when the intersection over union (IOU) threshold is set to 0.5 and 0.75, respectively. A larger threshold indicates more accurate predicted object locations. The baseline method refers to detection using multilevel features extracted from the CSPNeXt-m backbone, with the dimensions adjusted to 256.

**Table 1.** Ablation experiments on the DIOR dataset.

| Method | $mAP_{50}$ | $mAP_{75}$ | FPS | Parameters (M) |
| --- | --- | --- | --- | --- |
| Baseline | 74.2 | 58.2 | 34.1 | 15.0 |
| Baseline + ALFFE | 76.5 (**+2.1**) | 60.1 (**+1.9**) | 28.0 | 17.9 |
| Baseline + ALFFE + ASFF | 77.1 (**+2.9**) | 60.8 (**+2.6**) | 26.6 | 21.7 |

**Table 2.** Ablation experiments on the HRRSD dataset.

| Method | $mAP_{50}$ | $mAP_{75}$ |
| --- | --- | --- |
| Baseline | 88.0 | 77.0 |
| Baseline + ALFFE | 88.6 (**+0.6**) | 79.8 (**+2.8**) |
| Baseline + ALFFE + ASFF | 88.9 (**+0.9**) | 79.0 (**+2.0**) |

The mAP, FPS, and the number of parameters achieved by the different methods on the DIOR dataset are shown in Table 1 (where FPS and parameters are obtained at the input image size of 800 × 800). The baseline method's $mAP_{50}$ and $mAP_{75}$ achieve 74.2% and 58.2%, respectively. The $mAP_{50}$ and $mAP_{75}$ can be improved by 2.1% and 1.9%, respectively,

to 76.5% and 60.1% with the addition of the ALFFE designed in this paper. For example, in Figure 5(a1–a4,b1–b4), the addition of ALFFE to the baseline can avoid false detection of vehicles and accurately locate the bridge. The addition of ASFF further improves the $mAP_{50}$ and $mAP_{75}$ by 77.1% and 60.8%, respectively, since it can adaptively fuse features of different scales to improve the scale invariance of the features and better adapt to objects of different scales. At the same time, Figure 5(c1–c4) amply demonstrate that ALFFE can equip the model with the capability to accurately extract the discriminative features required to identify a class of objects, and ASFF can achieve more accurate object localization through the integration of different levels of features. Compared to the ground truth, although the localization of the detected chimney is not especially accurate, it is still a satisfactory detection result, since accurate identification and localization is extremely challenging in the case of significant intra-class variation. Furthermore, our method achieves high accuracy while the number of parameters in the model is small and the detection speed is moderate.

As observed in Table 2, the baseline method reaches 88.0% for $mAP_{50}$ on the HRRSD dataset, improving to 88.6% with the addition of ALFFE, and then 88.9% with the addition of ASFF. The $mAP_{75}$ improvement is more pronounced compared to $mAP_{50}$, since ALFFE and ASFF enable the model to locate objects and predict their categories more accurately.

### 4.4. Quantitative Comparison and Analysis

To assess the performance of the proposed method, comparative experiments are conducted on three commonly used remote sensing image datasets. The methods used for comparison on the DIOR dataset include the two-stage methods CSFF [35], FRPNet [33], CANet [28], and MFPNet [43]; the one-stage anchor-based methods HawkNet [44], and ASSD [45]; and the anchor-free methods O$^2$-DNet [46], and AFDet [47]. Table 3 shows the detection results of the different methods on the DIOR dataset. Compared with other methods, our method achieves the best mAP with smaller computational effort. In addition, the performance of the proposed method is the best for the classes of airplane, ship, vehicle, etc., indicating that the proposed method has satisfactory detection results for small objects.

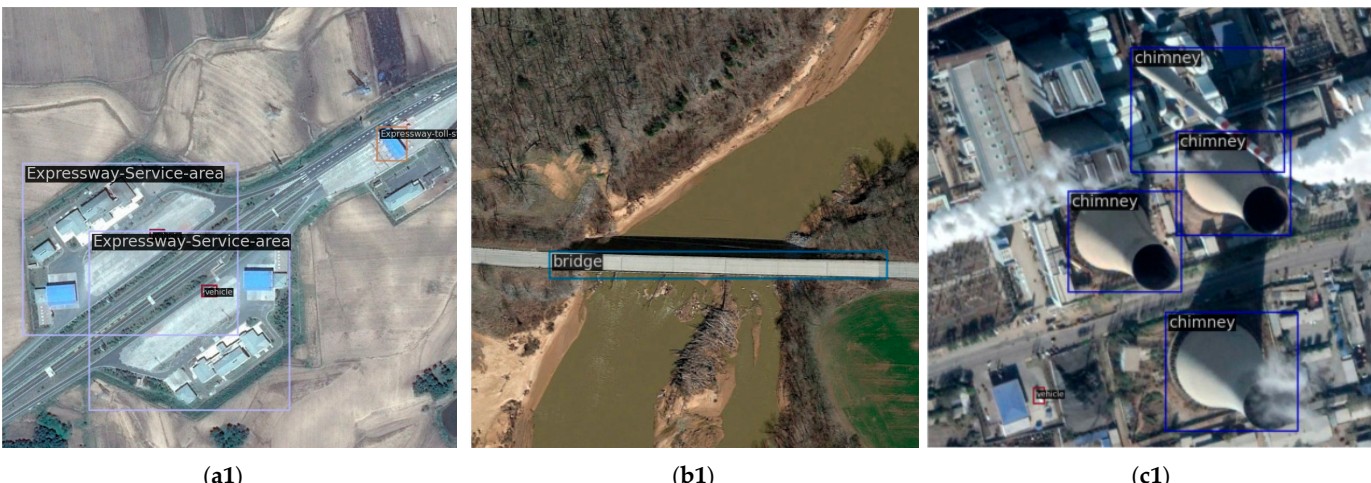

|      (a1)      |      (b1)      |      (c1)      |

**Figure 5.** *Cont.*

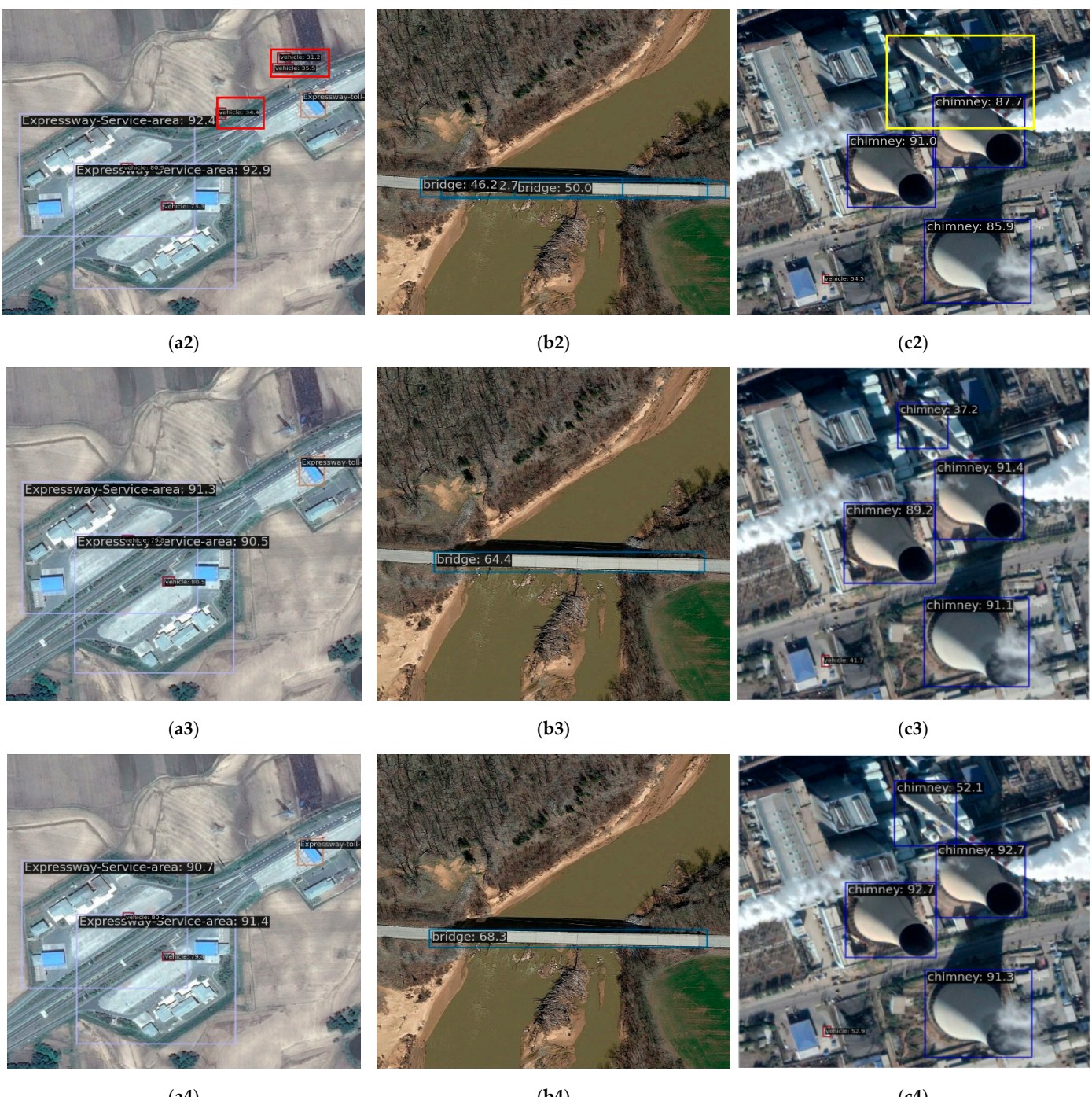

**Figure 5.** Visual comparisons of ablation experiment on the DIOR dataset: (**a1,b1,c1**) denote ground truth; (**a2,b2,c2**) denote the detection results of baseline; (**a3,b3,c3**) denote the detection results of baseline + ALFFE; (**a4,b4,c4**) denote the detection results of baseline + ALFFE + ASFF (red boxes indicate false detections and the yellow box indicates a missing detection).

The performance of different methods on the HRRSD dataset is summarized in Table 4, which includes HRCNN [39], Lang et al. [48], FCOS [23], and RepPoints [49]. It can be noticed that the proposed method achieves the best mAP using a backbone of moderate complexity. Moreover, the best AP is achieved for some categories such as basketball court and T junction, suggesting that the proposed method may excel at capturing the texture features inside the objects.

**Table 3.** Quantitative results of different object detection methods on the DIOR dataset. (Param means the number of model parameters).

| Method | AP | AI | BD | BC | BR | CH | DA | ESA | ETS | GF | GTF | HA | OV | SH | SD | ST | TC | TS | VE | WM | mAP | Param |
|--------|----|----|----|----|----|----|----|-----|-----|----|-----|----|----|----|----|----|----|----|----|----|-----|-------|
| MFPNet | 76.6 | 83.4 | 80.6 | 82.1 | 44.3 | 75.6 | 68.5 | 85.9 | 63.9 | 77.3 | 77.2 | 62.1 | 58.8 | 77.2 | 76.8 | 60.3 | 86.4 | 64.5 | 41.5 | 80.2 | 71.2 | / |
| O$^2$-DNet | 61.2 | 80.1 | 73.7 | 81.4 | 45.2 | 75.8 | 64.8 | 81.2 | 76.5 | 79.5 | 79.7 | 47.2 | 59.3 | 72.6 | 70.5 | 53.7 | 82.6 | 55.9 | 49.1 | 77.8 | 68.4 | / |
| ASSD | 85.6 | 82.4 | 75.8 | 80.5 | 40.7 | 77.6 | 64.7 | 67.1 | 61.7 | 80.8 | 78.6 | 62.0 | 58.0 | 84.9 | 65.3 | 65.3 | 87.9 | 62.4 | 44.5 | 76.3 | 71.1 | / |
| HawkNet | 65.7 | 84.2 | 76.1 | 87.4 | 45.3 | 79 | 64.5 | 82.8 | 72.4 | 82.5 | 74.7 | 50.2 | 59.6 | 89.7 | 66 | 70.8 | 87.2 | 61.4 | 52.8 | 88.2 | 72.0 | / |
| CSFF | 57.2 | 79.6 | 70.1 | 87.4 | 46.1 | 76.6 | 62.7 | 82.6 | 73.2 | 78.2 | 81.6 | 50.7 | 59.5 | 73.3 | 63.4 | 58.5 | 85.9 | 61.9 | 42.9 | 86.9 | 68.0 | / |
| AFDet | 82.4 | 81.5 | 81.9 | 89.8 | 51.7 | 74.9 | 58.7 | 84.2 | 73.3 | 79.5 | 81.0 | 44.2 | 62.0 | 77.8 | 63.2 | 76.9 | 91.0 | 62.5 | 59.3 | 87.1 | 73.2 | 20.29 |
| FRPNet | 64.5 | 82.6 | 77.7 | 81.7 | 47.1 | 69.6 | 50.6 | 80.0 | 71.7 | 81.3 | 77.4 | 78.7 | 82.4 | 62.9 | 72.6 | 67.6 | 81.2 | 65.2 | 52.7 | 89.1 | 71.8 | / |
| CANet | 70.3 | 82.4 | 72.0 | 87.8 | 55.7 | 79.9 | 67.7 | 83.5 | 77.2 | 77.3 | 83.6 | 56.0 | 63.6 | 81.0 | 79.8 | 70.8 | 88.2 | 67.6 | 51.2 | 89.6 | 74.3 | 64.60 |
| Ours | **90.0** | 80.5 | 81.0 | **90.1** | 52.5 | 78.7 | 65.7 | 84.0 | 73.5 | 81.2 | 80.7 | 65.3 | 65.4 | 88.9 | **84.7** | **78.6** | 89.9 | **68.3** | 62.3 | 80.1 | **77.1** | 21.70 |

**Table 4.** Quantitative results of different object detection methods on the HRRSD dataset ("Dark-tiny": CSPDarkNet-tiny).

| Method | Backbone | AP | BD | BC | BR | CR | GTF | HA | PL | SH | ST | TJ | TC | VE | mAP |
|--------|----------|----|----|----|----|----|-----|----|----|----|----|----|----|----|-----|
| HRCNN | ResNet-101 | 82.93 | 72.11 | 24.94 | 28.31 | 32.26 | 80.57 | 61.57 | 21.35 | 57.64 | 78.76 | 10.25 | 74.83 | 42.84 | 51.43 |
| FCOS | ResNet-101 | 96.82 | 91.21 | 54.10 | 89.69 | 94.42 | 97.45 | 95.05 | 63.15 | 90.46 | 94.91 | 82.23 | 87.85 | 91.82 | 86.86 |
| Lang et al. | Dark-tiny | 98.94 | 91.61 | 71.30 | 85.31 | 89.98 | 95.89 | 92.11 | 60.79 | 89.72 | 97.28 | 73.15 | 93.61 | 93.43 | 87.16 |
| RepPoints | ResNet-101 | 97.49 | 91.90 | 60.83 | 91.04 | 95.10 | 98.22 | 95.70 | 71.78 | 90.05 | 94.36 | 82.26 | 89.83 | 94.25 | 88.71 |
| Ours | CSPNeXt-m | 97.10 | 89.80 | **84.20** | 90.40 | 89.70 | 90.80 | 90.10 | **77.30** | 90.40 | 90.50 | **84.60** | 90.50 | 90.50 | **88.90** |

Table 5 shows the performance of different methods on the NWPU VHR-10 dataset, comparing methods such as ABNet [50], CANet [28], MSGNet [51], EVCP [52], and Liu et al. [53]. The proposed method achieves the highest detection accuracy. These results highlight the effectiveness and generality of the proposed method for multi-category remote sensing image object detection tasks.

**Table 5.** Quantitative results of different object detection methods on the NWPU VHR-10 dataset.

| Method | AP | BC | BR | GTF | HA | SH | ST | TC | VE | BF | mAP |
|---|---|---|---|---|---|---|---|---|---|---|---|
| ABNet | 100 | 95.98 | 69.04 | 99.86 | 94.26 | 92.58 | 97.77 | 99.26 | 95.62 | 97.76 | 94.21 |
| CANet | 100 | 90.60 | 93.90 | 99.80 | 89.80 | 81.90 | 94.60 | 90.70 | 89.90 | 90.30 | 92.20 |
| MSGNet | 92.93 | 92.02 | 91.07 | 99.98 | 99.09 | 93.68 | 97.90 | 91.82 | 92.22 | 98.60 | 95.53 |
| EVCP | 98.90 | 91.60 | 87.80 | 99.70 | 91.80 | 92.50 | 99.80 | 91.10 | 88.60 | 99.80 | 94.10 |
| Liu et al. | 99.50 | 95.40 | 82.20 | 99.20 | 89.60 | 88.40 | 90.20 | 89.20 | 92.90 | 98.70 | 92.50 |
| Ours | 99.60 | 92.90 | **96.30** | 99.10 | 97.20 | 91.40 | **100** | 90.30 | 90.60 | 99.30 | **95.70** |

*4.5. Visualization and Analysis*

Some typical detection results on the DIOR dataset are shown in Figure 6. It can be noticed that the proposed method obtains satisfactory detection results for objects with complex backgrounds, large variations in object scales, and high inter-class similarity. The proposed method can alleviate the problem of small object information loss and has the capability to extract discriminative features from objects, such as the detection of the airplanes in Figure 6a, the ships in Figure 6b, and the tennis court in Figure 6h. In addition, our method exhibits the ability to accurately extract object information from complex backgrounds, such as the ground track field in a complex background in Figure 6g and an airport with an irregular aspect ratio in Figure 6f. Moreover, the accurate detections of the overpass in Figure 6c and the bridge in Figure 6d demonstrates that our method has fine contextual information extraction capability, as the bridge and overpass have similar features and contextual information is required to distinguish them well.

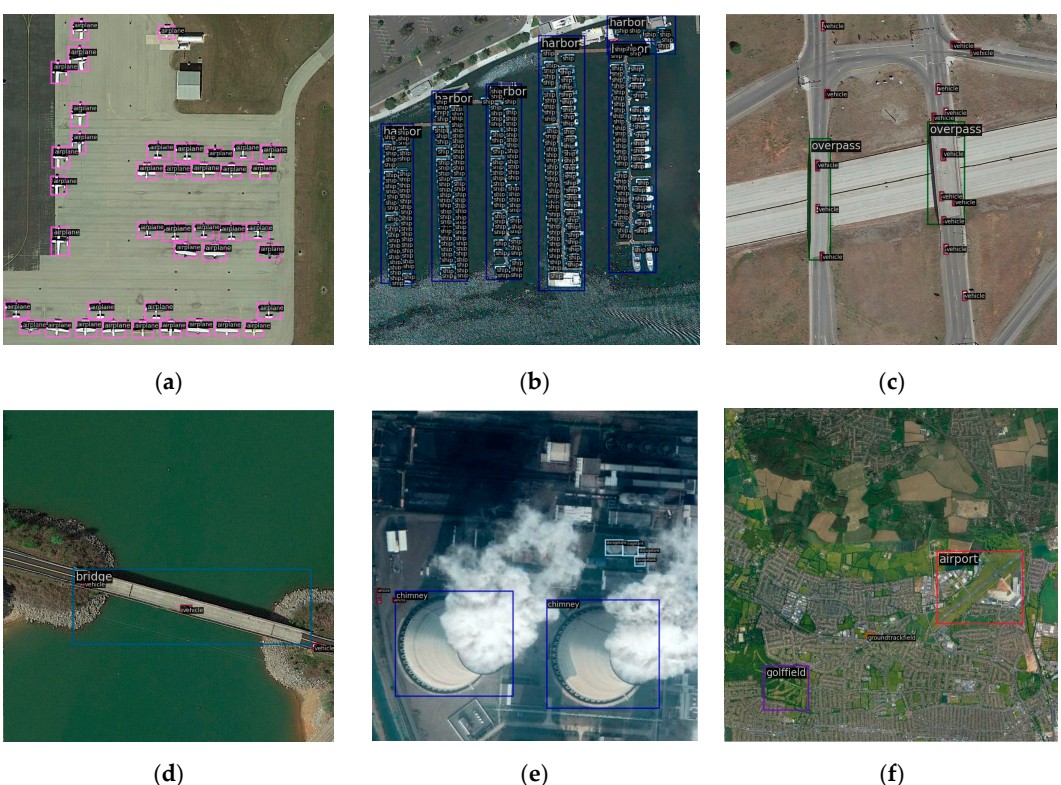

(**a**)  (**b**)  (**c**)

(**d**)  (**e**)  (**f**)

**Figure 6.** *Cont.*

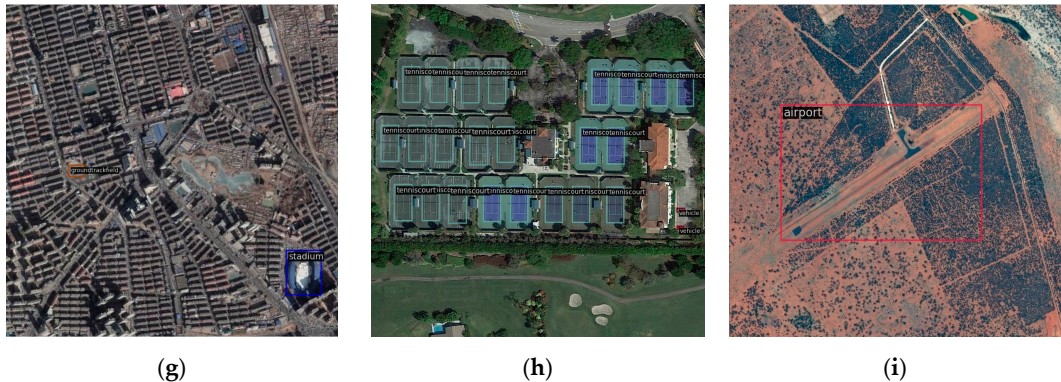

**Figure 6.** Detection results of different objects on the DIOR dataset. (**a**,**b**,**e**,**h**) Objects at different scales; (**c**,**d**) objects with high inter-class similarity; (**f**,**g**,**i**), objects in a complex background.

Remote sensing images exhibit variations in imaging conditions and image quality, posing challenges for object detection. In Figure 7, challenging scenes with clouds, uneven chromaticity, color distortion, low resolution, and shadowing are depicted. The proposed method demonstrates its effectiveness in accurately extracting discriminative features and detecting objects in such scenes, where the semantic information of objects is difficult to capture and can be easily suppressed by noise.

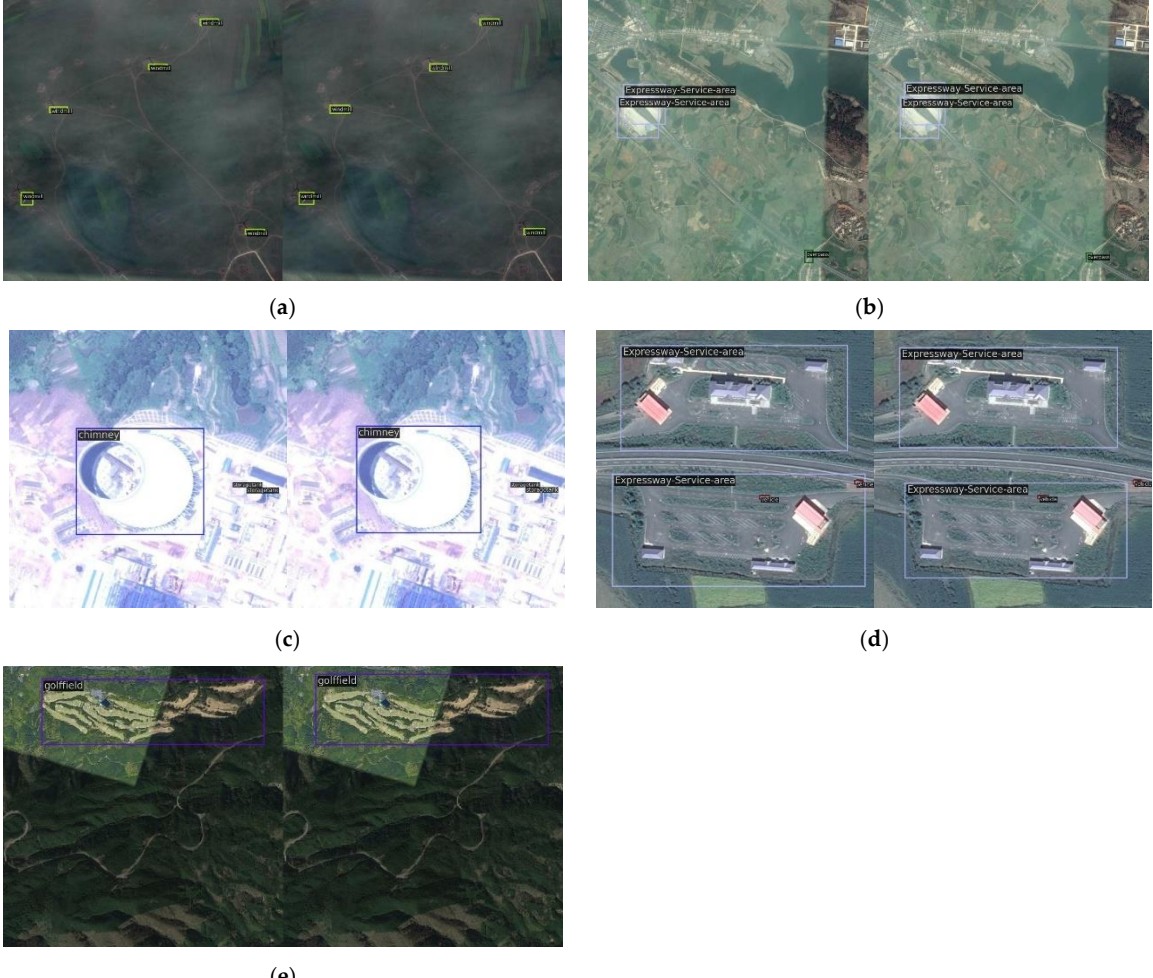

**Figure 7.** Detection results of challenging conditions on the DIOR dataset. (**a**) Clouds; (**b**) uneven chromaticity; (**c**) color distortion; (**d**) low resolution; (**e**) shadowing. (**a**–**e**) Figures are stitched images where the left images are the ground truth and the right are the detection results.

The detection results of some typical objects in HRRSD dataset are shown in Figure 8.

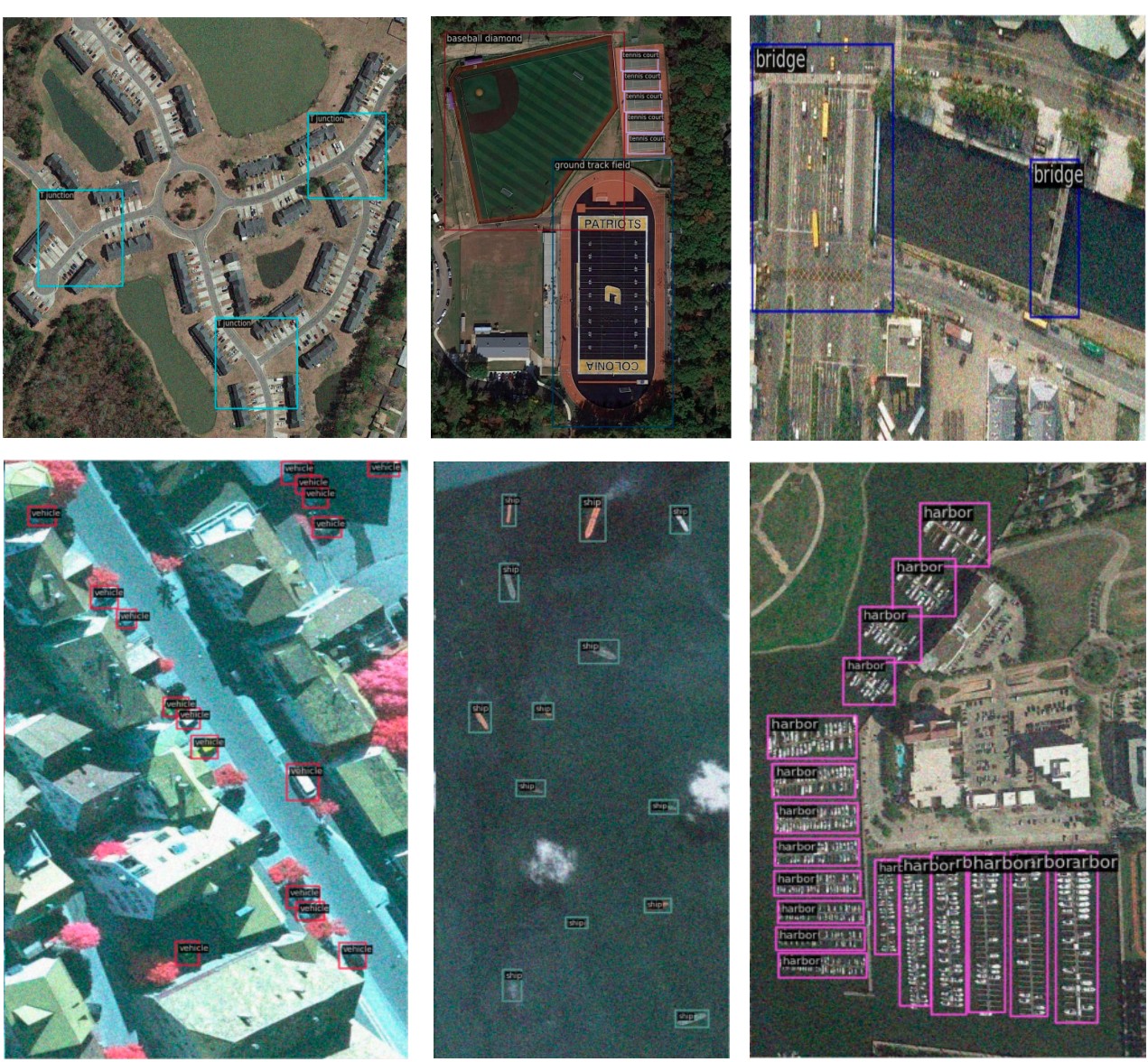

**Figure 8.** Detection results of different objects on the HRRSD dataset.

In the results obtained on the NWPU VHR-10 dataset, the feature maps used for prediction in the model are visualized using AblationCAM [54] to visualize the perceptual capability of the proposed method for the objects. The results of some typical objects are shown in Figure 9. In this case, the shift from blue to red in the figure indicates the level of weak-to-strong focus on this area. The warm color means that the model is paying more attention to this area. It can be seen from Figure 9 that the method proposed can accurately focus on the centers of all types of objects, almost unaffected by the surrounding environment.

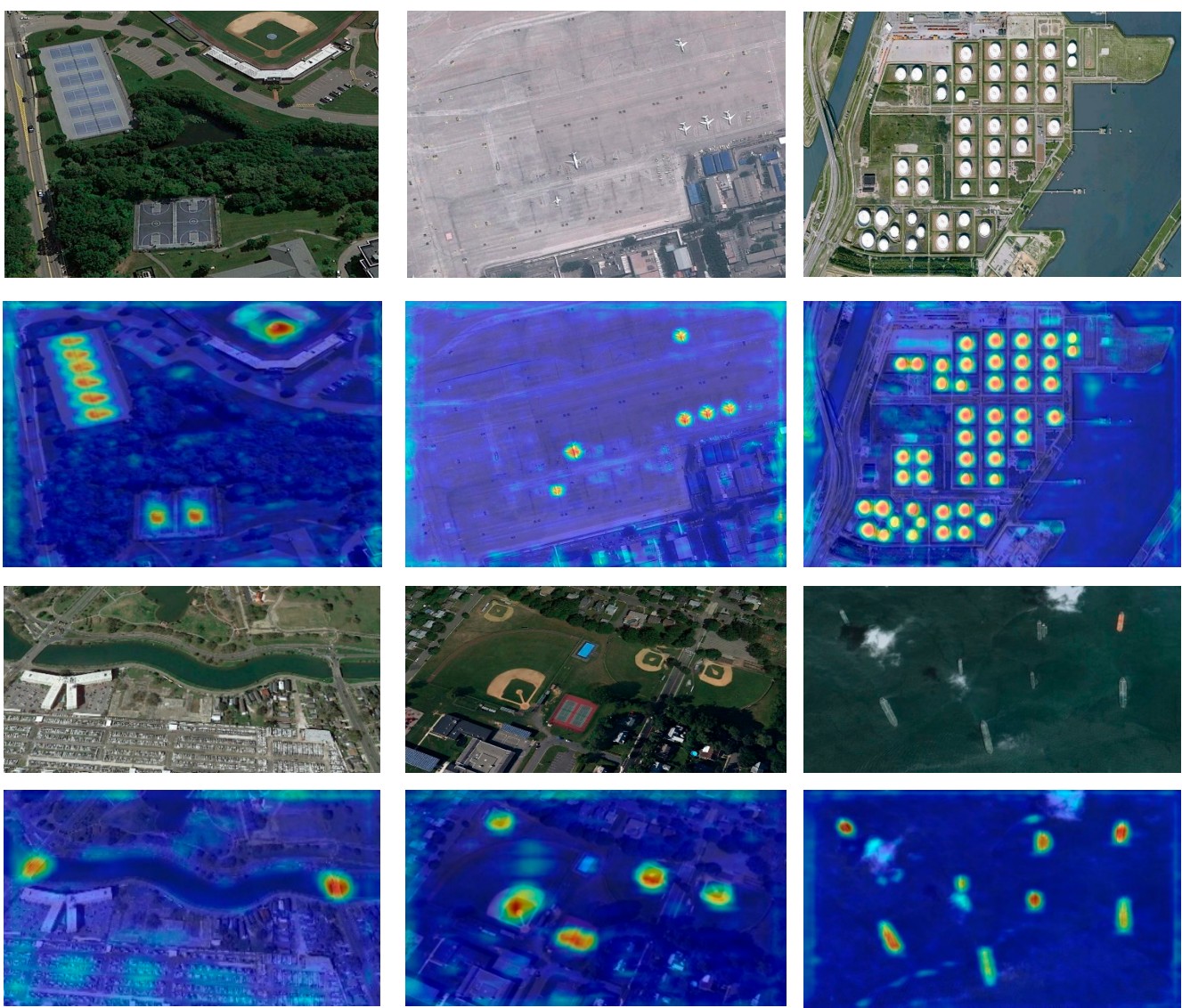

**Figure 9.** Visualization of the regions of interest detected by the model.

## 5. Conclusions

This paper aims to address the challenges caused by the large variation in object scales, complex environmental backgrounds, and high inter-class similarity in remote sensing image object detection. We propose an effective adaptive adjacent layer feature fusion (AALFF) method for remote sensing image object detection based on RTMDet. The major contributions include: (1) Designing an adjacent layer feature fusion enhancement module (ALFFE). (2) Introducing the adaptive spatial feature fusion module (ASFF). The ALFFE module enables the model to better capture high-level semantic information of objects and accurately locate their spatial positions by fusing adjacent layer features. It also mitigates the problem of missed small objects and enhances the capture of crucial object information within complex backgrounds. The ASFF is introduced to adaptively adjust the degree of feature participation at different scales during fusion to better accommodate objects with varying scales. Ablation experiments on the DIOR and HRRSD datasets demonstrate the effectiveness of different components in AALFF. Our method achieves state-of-the-art performance on the DIOR dataset. Furthermore, our method achieves satisfactory performance with a small number of model parameters, striking a good balance between speed and accuracy. Experimental results on the DIOR, HRRSD, and NWPU VHR-10



datasets demonstrate the successful detection of objects with large scale variation, complex backgrounds, and high inter-class similarity.

Currently, deep-learning-based methods for remote sensing image object detection often require a large number of annotated samples for training deep neural networks. However, acquiring a large-scale annotated dataset in the remote sensing field is a challenging task. Additionally, with the continuous advancement of remote sensing technology, new remote sensing scenarios and object types are continuously emerging. Detecting these new scenarios and objects often faces the challenge of limited data availability. Therefore, in the future, our research will focus on few-shot object detection in remote sensing images, aiming to effectively utilize limited data resources and enhance the effectiveness and accuracy of object detection.

**Author Contributions:** Conceptualization, Z.G.; methodology, X.Z.; software, X.L. and K.Z.; formal analysis, H.G. and J.W.; writing—original draft preparation, X.Z.; writing—review and editing, H.G. and L.D.; funding acquisition, L.D. All authors have read and agreed to the published version of the manuscript.

**Funding:** This research was funded by the National Science Foundation of China, grant number 42201443.

**Conflicts of Interest:** The authors declare no conflict of interest.

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
