# Peer review of "Adaptive Adjacent Layer Feature Fusion for Object Detection in Remote Sensing Images"

_remotesensing, doi:10.3390/rs15174224_

Round 1

Reviewer 1 Report

Dear sir 

The paper is technically sound with no further comments.

Good Luck

Author Response

We sincerely appreciate the time and effort you have dedicated to reviewing our manuscript. 

Reviewer 2 Report

Ok about your work. However, I have 2 questions:

1) why not use the same methods for different data sets? By using different methods for each dataset, it gives the impression that you have only chosen the methods with a lower result than your method. You need to explain the reason for these different choices or use all the methods in each dataset. 

2) When you say  in 4.4: "with smaller computational effort", could you quantifie this effort?

Author Response

Thank you for your meticulous review and insightful comments on our manuscript. Your expertise and attention to detail have greatly contributed to the improvement of our research. And we are grateful for the time and effort you have dedicated to reviewing our work. 

Please find our point-by-point response to your comments in attachment.

Reviewer 3 Report

This paper proposed Adaptive Adjacent Layer Feature Fusion for Object Detection in Remote Sensing Images. Overall, the structure of this paper is well organized, and the presentation is clear. However, there are still some crucial problems that need to be carefully addressed before a possible publication. More specifically,

1.     The motivations or remaining challenges are not so clear or what kinds of issues or difficulties are this task that is facing. Please give more details and discussion about the key problems solved in this paper, which is largely different from existing works.

2.     A deep literature review should be given, particularly regarding currently advanced deep learning classification techniques and algorithms in remote sensing. As a result, the reviewer suggests discussing some advanced works in the revised manuscript, e.g., 10.1109/TGRS.2020.3016820, 10.1109/TGRS.2021.3130716, 10.1109/TGRS.2020.3015157.

3.     The contributions of this paper are not so clear to the reviewer. Please clarify which one is existing and which one is your own?

4.     The ablation analysis should be given.

5.     The reviewer is also wondering the computational complexity of the proposed method.

6.     Some SOTA object detection in remote sensing should be added for discussion, e.g., ORSIm detector, UIU-Net.

7.     Some future directions should be pointed out in the conclusion.

No more.

Author Response

Thank you for your meticulous review and insightful comments on our manuscript. Your expertise and attention to detail have greatly contributed to the improvement of our research. We have carefully considered each of your suggestions and made the necessary revisions to address the concerns raised. Your expertise and meticulous evaluation have greatly benefited our research, and we are grateful for the time and effort you have dedicated to reviewing our work. 

Please find our point-by-point response to your comments and the revised manuscript (with the modifications highlighted in red) in attachment.

Round 2

Reviewer 2 Report

Ok, thanks for  your answers

Reviewer 3 Report

No more comments.